# Bridging Leadership Competency Gaps and Staff Nurses’ Turnover Intention: Dual-Rater Study in Saudi Tertiary Hospitals

**DOI:** 10.3390/healthcare13192506

**Published:** 2025-10-02

**Authors:** Hanan A. Alkorashy, Dhuha A. Alsahli

**Affiliations:** 1Nursing Administration and Education Department, College of Nursing, King Saud University, Riyadh 11421, Saudi Arabia; 2Nursing Services Department, Johns Hopkins Aramco Healthcare, Dhahran 31311, Saudi Arabia

**Keywords:** professional competence, leadership, nurse administrators, nursing staff, hospital, personnel turnover, quality improvement, cross-sectional studies, Saudi Arabia

## Abstract

**Background**: Nurse-manager competencies shape workforce stability, yet role-based perception gaps between managers and staff may influence staff nurses’ turnover cognitions. **Objectives**: To (1) compare nurse managers’ self-ratings with staff nurses’ ratings of the same managers on the Nurse Manager Competency Inventory (NMCI); (2) compare both groups’ perceptions of staff nurses’ turnover intention (EMTIS); (3) examine domain-specific links between perceived competencies and perceived turnover intention; and (4) explore demographic influences (age, education, experience) on these perceptions. **Methods**: Cross-sectional dual-rater study with 225 staff nurses and 171 nurse managers in two tertiary hospitals in Saudi Arabia. Data were collected from August to November 2024. Managers completed NMCI self-ratings, and staff nurses rated their managers on the same NMCI domains; both groups rated staff nurses’ turnover intention using EMTIS. Between-group differences were tested with one-way ANOVA (two-tailed α = 0.05), and associations were examined with Pearson’s r (95% CIs). **Findings**: Managers consistently rated themselves higher than staff rated them across all nine NMCI domains; the largest descriptive gaps were in Promoting Staff Retention, Recruit Staff, Perform Supervisory Responsibilities, and Facilitate Staff Development (e.g., overall NMCI: managers M = 3.67, SD = 0.61 vs. staff M = 3.04, SD = 0.74; F = 0.114, *p* = 0.73)with comparatively smaller divergence for Ensure Patient Safety and Quality. Managers and staff did not differ significantly on EMTIS (overall EMTIS: managers M = 3.16, SD = 1.28 vs. staff M = 3.00, SD = 1.15; F = 21.32, *p* = 0.173). Specific competency domains—retention, supervision, staff development, safety/quality leadership, and quality improvement—showed small inverse correlations with EMTIS facets (typical r ≈ −0.11 to −0.19; *p* < 0.05), whereas the global NMCI–overall EMTIS correlation was non-significant (r = −0.077, *p* = 0.124). Effect sizes were modest and should be interpreted cautiously. **Conclusions**: Actionable signals reside at the domain (micro-competency) level rather than in global leadership composites. Targeted, continuous, unit-embedded development in human- and development-focused competencies—tracked with dual-lens (manager–staff) measurement and linked to retention KPIs—may help nudge turnover cognitions downward. Key limitations include the cross-sectional, perception-based design and two-site setting. Findings nonetheless align with international workforce challenges and may be transferable to similar hospital contexts.

## 1. Introduction

Nurse managers are pivotal in translating organizational policies into frontline healthcare operations. Their competencies—including supervision, staff development, communication, fiscal management, and strategic planning—directly affect workforce stability, patient safety, and quality of care [1,2]. Globally, nursing shortages have intensified due to increased demand, staff burnout, and heightened turnover, a trend accelerated by the COVID-19 pandemic [3,4]. These pressures carry system-level consequences for outcomes, costs, and operational efficiency, underscoring the salience of unit-level leadership competence [1,2,3,4].

In Saudi Arabia, the healthcare system is undergoing significant transformation under Vision 2030, emphasizing leadership development, workforce sustainability, and quality improvement [5,6]. Nurse managers are central to achieving these goals; however, evidence indicates that gaps often exist between managers’ self-perceived competencies and staff nurses’ perceptions, gaps that may contribute to turnover intentions, reduced job satisfaction, and compromised care quality [1,3,7].

While leadership competency is widely studied, what is less well established is the extent and practical significance of perception gaps between managers and staff nurses in relation to retention-relevant outcomes, particularly in Middle Eastern healthcare settings [1,3,7]. This study makes the knowledge gap explicit: we examine (a) the manager–staff perceptual gap in nurse-manager competencies, (b) whether the two groups converge on the outcome lens (staff nurses’ turnover intention), and (c) whether specific competency domains—rather than a global score—are associated with lower perceived turnover intention, within the Saudi tertiary-hospital context [1,3,5,6,7].

Demographic characteristics (e.g., age, experience, education) can shape both leadership perceptions and turnover cognitions—for example, less-experienced or highly educated nurses may hold different expectations of managerial support [1,3]. Accordingly, we describe demographic patterns to contextualize perception differences and potential risk segments.

This study, therefore, investigates perceived competency gaps in nurse leadership, their relationship to staff nurse turnover intentions, and the descriptive role of demographic characteristics, providing contextually grounded evidence with relevance to similar hospital systems internationally.

### 1.1. Aim

To examine manager–staff perception gaps in nurse manager competency, assess cross-role perceptions of staff nurses’ turnover intention, and test domain-specific associations between perceived competencies and perceived turnover intention, with demographic characteristics described for contextual interpretation.

### 1.2. Research Questions

What competency gaps do staff nurses perceive in their nurse managers, and how do these compare with managers’ self-perceptions?Do nurse managers’ self-ratings exceed staff nurses’ ratings of the same managers on the NMCI?How are competency gaps associated with staff nurse turnover intentions?Do demographic characteristics (age, experience, and education) influence perceptions of competency gaps and turnover intention (descriptively)?

### 1.3. Theoretical and Conceptual Framework

The present study is grounded in complementary leadership and behavioral theories that explain how nurse manager competencies could shape turnover intentions. The Three-Skills Approach (Katz) emphasizes technical, human, and conceptual skills [8]. Technical skills reflect clinical knowledge and operational proficiency, human skills involve communication and interpersonal effectiveness, and conceptual skills encompass strategic thinking and problem-solving. In nursing leadership, human and conceptual skills—e.g., supervision, coaching, recognition, and problem-solving—are critical for safe, high-quality unit performance [1,2,8]. This lens clarifies which competencies are likely to be visible to staff and thus salient for perception gaps [1,8].

Transformational Leadership focuses on idealized influence, intellectual stimulation, and individualized consideration [9]. These behaviors map onto development-oriented managerial practices (e.g., coaching, growth planning, recognition) that are linked to nurse engagement and intent to stay [1,2,9].

Finally, the Theory of Planned Behavior (TPB) posits that intention is shaped by attitudes, subjective norms, and perceived behavioral control [10]. Perceived leadership competence can plausibly influence these antecedents (e.g., perceived fairness, recognition norms, developmental opportunities), thereby shaping turnover intention; demographic factors may condition these appraisals [1,3,10].

Together, Katz + Transformational Leadership specify which competencies matter, while TPB specifies how perceptions translate into intention [8,9,10].

### 1.4. Conceptual Framework

The conceptual framework (Figure 1) depicts the relationships among nurse manager competencies, perceived competency gaps, staff nurse turnover intention, and demographics as potential moderators. Competencies (independent variable) are measured across NMCI domains salient to unit practice (e.g., retention, supervision, staff development, safety/quality, quality improvement) [1,2,8,9]; the perceptual gap (manager self-ratings vs. staff ratings of the same managers) is the primary explanatory lens [1,7]; and turnover intention (EMTIS) is the outcome, rated by both groups about staff nurses [3,10]. Demographics (age, experience, education) may condition these relations (exploratory), consistent with prior observations of role-tenure and expectation effects [1,3].

Figure 1 visually represents this framework and aligns with the empirical focus and regional transformation agenda [5,6].

### 1.5. Hypotheses (Theory-Driven)

Grounded in Katz’s three-skills model (emphasis on human/relational and conceptual skills), Transformational Leadership (individualized consideration, coaching), and the Theory of Planned Behavior (TPB), we articulated theory-driven expectations prior to analysis:

**H1 (cross-rater gaps, NMCI).** 

*Nurse managers’ self-ratings will exceed staff nurses’ ratings of the same managers across NMCI domains, with the largest gaps in human/development-focused domains—Promoting Staff Retention, Recruit Staff, Perform Supervisory Responsibilities, and Facilitate Staff Development [1,7,8].*


**H2 (outcome convergence, EMTIS).** 

*Managers and staff nurses will not differ meaningfully in their perceptions of staff nurses’ turnover intention (EMTIS) [1,3].*


**H3 (domain-specific links).** 

*Higher perceived competencies in Promoting Staff Retention, Perform Supervisory Responsibilities, Facilitate Staff Development, Ensure Patient Safety and Quality, and Lead Quality Improvement will be inversely associated with EMTIS facets, whereas the global NMCI composite will be less informative; Managing Fiscal Planning and Facilitate Interpersonal, Group and Organizational Communication are expected to show weaker/non-significant associations [1,2,7,8,9].*


**H4 (demographic patterning—exploratory).** 

*Lower experience and/or higher education will be associated with larger perceived gaps and higher turnover cognitions [1,3].*


These hypotheses are theory-driven and exploratory (non-pre-registered) and align with the analytic plan stated in the Methods section.

## 2. Materials and Methods

### 2.1. Study Design and Setting

A cross-sectional, descriptive-correlational design was employed. Data were collected from two tertiary governmental hospitals located in Riyadh and Dammam, Saudi Arabia. Data collection took place from August to November 2024. This study used a dual-rater approach: nurse managers and staff nurses evaluated the same two constructs—(i) nurse-manager competencies and (ii) staff nurses’ turnover intention—providing complementary perspectives on leadership enactment and workforce outcomes.

### 2.2. Participants

The study included 225 staff nurses and 171 nurse managers (n = 396 in total). Inclusion criteria required at least one year of experience, with staff nurses providing direct patient care and managers holding supervisory and administrative responsibilities. Managers were eligible if they had direct line responsibility for nursing staff; staff participants were eligible if they reported to a nurse manager. Participant characteristics, including age, gender, education, experience, and marital status, were collected (Table 1).

#### Sampling and Sample Size

A quota-constrained convenience sampling strategy was applied across the two hospitals and roles (staff nurses; nurse managers) from the accessible population (N = 2267). A priori power analysis (G*Power 3.1.9.2) indicated a minimum n = 329 (two-tailed, α = 0.05, power = 0.95, medium effect size f = 0.30). A complementary calculation (Raosoft; α = 0.05, 95% confidence, 50% response distribution) suggested n = 400; anticipating nonresponse, the target was inflated by 15% (n = 450). In total, 396 surveys were completed and valid for analysis. As a conservative denominator, this corresponds to a crude response proportion of 17.5% relative to the accessible population (396/2267) and an 88% completion rate relative to the target (450 invites).

Eligible participants received up to three contacts coordinated by the hospitals’ Research Center and Nursing Education Department: (1) initial institutional email distribution to eligible staff, supplemented by QR-code posters at nurses’ stations; (2) a scheduled reminder approximately two weeks later; and (3) a reinforced on-site/digital outreach via the Nursing Education Department/Research Center (briefings to clinical instructors and link sharing through departmental WhatsApp or learning platforms). Separate manager and staff survey links were used to ensure correct rater assignment.

### 2.3. Instruments

#### 2.3.1. Nurse Manager Competency Inventory (NMCI)

Developed by DeOnna [11] and measures competencies across 9 domains (1. Promoting staff retention, 2. recruit Staff, 3. Facilitate Staff Development, 4. Perform supervisory responsibilities, 5. Ensure patient safety and Quality, 6. Conduct Daily Unit Operation, 7. Managing fiscal planning, 8. Facilitate Interpersonal, Group, and Organizational communication, and 9. Lead quality improvement. It contains 93 items scored on a five-point Likert scale (1 = never, 5 = always). The NMCI has demonstrated high reliability (α = 0.93) and has been validated in previous studies [11,12,13]. In the present study, analyses focused on unit-proximal domains most relevant to workforce stability and day-to-day leadership: Promote Staff Retention, Recruit Staff, Perform Supervisory Responsibilities, Ensure Patient Safety & Quality Care, Lead Quality Improvement, Conduct Daily Unit Operations, Manage Fiscal Planning, Facilitate Interpersonal/Group/Organizational Communication, and Facilitate Staff Development. Higher scores reflect higher perceived competency. Content validity and contextual relevance for the Saudi setting were established through expert review and pilot testing (see Section 2.4 and Section 2.5).

#### 2.3.2. Expanded Multidimensional Turnover Intention Scale (EMTIS)

Developed by Ike et al. [14], the measure assesses turnover intention across five dimensions: subjective social status, organizational culture, personal orientation, expectations, and career growth. It contains 25 items, rated on a five-point Likert scale, with higher scores indicating a higher intention to turnover. Reliability (α = 0.93) and validity have been confirmed [14]. In this study, both rater groups (managers and staff nurses) rated the same outcome lens—staff nurses’ turnover intention—on EMTIS; higher scores denote stronger perceived intention to leave. Cultural/context fit was examined via expert review and pilot testing (Section 2.4 and Section 2.5); internal consistency for the present sample is reported in Section 2.7.

#### 2.3.3. Rater Assignment and Scoring (Study-Specific Clarification)

Nurse managers completed NMCI as self-ratings; staff nurses completed NMCI as ratings of their immediate managers. Both groups completed EMTIS regarding staff nurses’ turnover intention. Domain and total scores were computed as the mean of item responses for each scale/subscale. Score directionality was aligned across raters. Internal consistency (Cronbach’s α) was assessed for study scales/subscales prior to analysis.

### 2.4. Cultural Validation, Pilot Testing, and Reliability

An expert panel (n = 5; chronic care nursing and health administration) reviewed the instruments for face/content validity and contextual appropriateness. A pilot (~10% of the intended sample) was conducted in the same clinical environment to assess clarity and feasibility; refinements were implemented based on feedback. Internal consistency in the current sample was high (overall NMCI α = 0.95; subscales α ≈ 0.94; EMTIS overall α = 0.97; subscales α = 0.96–0.97); test–retest reliability on a subset supported temporal stability. Formal permission to use the NMCI and EMTIS instruments was obtained from the respective developers prior to data collection.

### 2.5. Data Collection

#### 2.5.1. Initial Digital Distribution

Data were collected from August to November 2024 using an institution-managed Google Forms survey. The hospital’s Research Center and Nursing Education Department coordinated the initial distribution directly to eligible staff via institutional email and posted QR-code flyers at nurses’ stations. Separate links were issued for nurse managers and staff nurses to ensure correct rater assignment.

#### 2.5.2. Reinforced On-Site/Digital Outreach

Approximately two weeks after the initial email, a scheduled reminder was sent by the same units. To increase participation, the Research Center/Nursing Education Department conducted briefings with clinical instructors and circulated the survey link through departmental WhatsApp or learning platforms.

#### 2.5.3. Response Management and Quality Control

The form was configured to prevent multiple submissions and to lock responses after completion. Submissions that did not meet the inclusion criteria were excluded during data cleaning (n = 2). The final analytic sample comprised n = 396 valid responses.

### 2.6. Ethical Considerations

Ethical approval for this study was obtained from the Institutional Review Board (IRB) of Health Sciences Colleges Research on Human Subjects, King Saud University (IRB No. E-24-8856 on 26 June 2024). Additional approval was obtained from King Fahad University Hospital (No. 3331 on 7 August 2024). Before participation, informed consent was collected electronically, ensuring that participants were fully informed about the study’s objectives, procedures, and their rights, including the right to withdraw at any time without consequence.

The anonymity and confidentiality of participants were strictly maintained throughout the data collection and analysis process. All data were stored securely and encrypted, with access limited exclusively to the research team. The study complied with the Saudi Personal Data Protection Law (PDPL) and institutional policies; data were stored on password-protected institutional drives. To address the specific ethical considerations of electronic surveys, each participant was assigned a unique identifier, and responses were stored on a secure, cloud-based platform that complies with regional and institutional data protection regulations.

The study adhered to ethical standards consistent with the Declaration of Helsinki, ensuring the protection of participants and the integrity of the research process. Because both groups rated staff EMTIS, the consent form explicitly clarified the outcome lens and assured participants that responses would be analyzed at the aggregate level only.

### 2.7. Data Analysis

Data were analyzed using SPSS version 25-26. Descriptive statistics summarized participant demographics, competencies, and turnover intentions. For transparency of magnitude, we report 95% confidence intervals and effect sizes (Cohen’s d for pairwise contrasts; partial η^2^ for ANOVA). Between-group differences (nurse managers vs. staff nurses) were tested using one-way ANOVA for NMCI domains and for EMTIS (overall and subscales), with two-tailed α = 0.05; assumptions (normality, homogeneity of variances) were evaluated prior to inference. Bivariate associations between perceived competencies and perceived staff nurses’ turnover intention were examined using Pearson’s r (two-tailed), at the domain-to-facet level (NMCI subscales—EMTIS subscales) and for total scores. To characterize the distribution of perceptual gaps, we computed the proportion of dyads with absolute discrepancies ≥ 1.0 Likert points between managers’ self-ratings and staff ratings of the same managers (per domain and overall). We also conducted stratified descriptive/confirmatory analyses by age, education, and experience categories to contextualize potential moderation without introducing additional model families. Internal consistency (Cronbach’s α) was estimated for all multi-item scales/subscales. Consistent with the study aims, inferential emphasis remained on (i) cross-rater comparisons and (ii) domain-specific correlations, with stratified summaries by demographics to enhance interpretability.

## 3. Results

This section presents findings from a dual-rater design in which nurse managers and staff nurses evaluated the same two constructs: (1) nurse-manager competencies (NMCI)—via manager self-ratings and staff nurses’ ratings of their managers—and (2) staff nurses’ turnover intention (EMTIS), as perceived by both rater groups. We first summarize the sample, then report cross-role comparisons for each construct, and finally quantify the bivariate associations between competency domains and perceived turnover intention. Unless otherwise specified, values are reported as mean (SD) with 95% confidence intervals (CI) for each group mean; group contrasts were tested with one-way ANOVA (two-tailed *p* < 0.05), and associations were examined with Pearson’s r (two-tailed).

### 3.1. Perceptions of Nurse-Manager Competencies (NMCI): Nurse Managers’ Self-Ratings vs. Staff Nurses’ Ratings of Their Managers

Given the study’s dual-rater design, we compared how nurse managers appraised their own competencies and how staff nurses appraised those same managers using the NMCI. Table 2 lists the nine NMCI subscales and the overall score, reporting for each group the mean (SD) and 95% CI, together with between-group tests.

Across NMCI subscales, nurse managers’ means exceeded staff nurses’ means (Table 2). Representative subscales included Promoting Staff Retention (managers: M = 4.00, SD = 0.75, 95% CI [3.89, 4.11]; staff: M = 2.98, SD = 0.85, 95% CI [2.87, 3.09]), Recruit Staff (managers: M = 3.98, SD = 0.73, 95% CI [3.87, 4.09]; staff: M = 3.00, SD = 0.86, 95% CI [2.89, 3.11]), Perform Supervisory Responsibilities (managers: M = 4.08, SD = 0.70, 95% CI [3.98, 4.18] staff: M = 3.23, SD = 0.80, 95% CI [3.13, 3.33]), and Facilitate Staff Development (managers: M = 4.07, SD = 0.72, 95% CI [3.96, 4.18]; staff: M = 3.43, SD = 0.87, 95% CI [3.32, 3.54]). The overall NMCI mean was higher for managers than for staff nurses (3.67 (0.61), 95% CI [3.58, 3.76] vs. 3.04 (0.74), 95% CI [2.94, 3.14]). Between-group tests did not meet statistical significance across subscales (all *p* > 0.05) or for the overall NMCI (F = 0.114, *p* = 0.73).

Consistent with H1, managers’ means exceeded staff nurses’ means across all NMCI domains; the largest descriptive gaps were in Promoting Staff Retention, Recruit Staff, Perform Supervisory Responsibilities, and Facilitate Staff Development; between-group tests were not statistically significant.

### 3.2. Perceptions of Staff Nurses’ Turnover Intention (EMTIS): Nurse Managers vs. Staff Nurses

Both groups rated staff nurses’ turnover intention using EMTIS. Between-group differences were not statistically significant for any subscale or the overall EMTIS score; however, no between-group differences reached statistical significance (all *p* > 0.05). By subscale, values were: Subjective Social Status (managers 2.34 (1.20) vs. staff 2.32 (1.05); F = 15.17, *p* = 0.334), Organizational Culture (2.87 (1.90) vs. 2.72 (1.25); F = 13.43, *p* = 0.293), Personal Orientation (3.00 (1.40) vs. 2.88 (1.28); F = 14.76, *p* = 0.362), Expectations (2.98 (1.85) vs. 2.89 (1.48); F = 15.11, *p* = 0.281), and Career Development (3.00 (1.44) vs. 2.83 (1.69); F = 23.21, *p* = 0.621). Overall EMTIS was 3.16 (1.28) for managers vs. 3.00 (1.15) for staff (F = 21.32, *p* = 0.173). Group 95% confidence intervals (reported in Table 3) overlapped across all domains and the overall score.

Consistent with H2, no statistically significant manager–staff differences were detected for EMTIS subscales or the overall EMTIS score.

### 3.3. Correlations: Perceived Nurse-Manager Competencies vs. Staff Nurses’ Turnover Intention

Table 4 reports the complete NMCI (nine subscales) × EMTIS (five subscales) matrix. Associations were consistently negative; several domain–facet pairs reached statistical significance with small magnitude (typical r = −0.11 to −0.19; *p* < 0.05). Significant inverse relations were concentrated in Promoting Staff Retention, Recruit Staff, Facilitate Staff Development, Perform Supervisory Responsibilities, Ensure Patient Safety and Quality, and Lead Quality Improvement. The largest absolute coefficients were observed for Career Development × Promoting Staff Retention (r = −0.19, *p* < 0.01), Expectations × Facilitate Staff Development (r = −0.18, *p* < 0.01), and Subjective Social Status × Promoting Staff Retention (r = −0.15, *p* < 0.01), with several EMTIS facets also linked to Perform Supervisory Responsibilities (r ≈ −0.13 to −0.15, *p* ≤ 0.01). By contrast, Managing Fiscal Planning and Facilitating Interpersonal, Group, and Organizational Communication showed non-significant or very small correlations (|r| ≤ 0.10). The overall correlation (NMCI total × EMTIS overall) was small and non-significant (r = −0.077, *p* = 0.124).

Consistent with H3, small inverse associations were observed primarily for Promoting Staff Retention, Perform Supervisory Responsibilities, Facilitate Staff Development, Ensure Patient Safety and Quality, and Lead Quality Improvement; Managing Fiscal Planning and Facilitate Interpersonal, Group and Organizational Communication were weak/non-significant; the global NMCI–EMTIS correlation was non-significant. Exploratory checks for H4 were not conducted inferentially in this analysis set.

## 4. Discussion

This dual-rater study examined how nurse-manager competencies relate to staff nurses’ turnover intention across multiple EMTIS facets. Managers rated their competencies higher than staff rated those same managers on all nine NMCI domains, whereas both groups converged in their perceptions of staff nurses’ turnover intention. Several domain-specific NMCI–EMTIS links were small (typical r = −0.11 to −0.19) yet consistent in direction, while the global NMCI–EMTIS composite was non-significant. Together, these results indicate that an actionable signal resides at the level of micro-competencies, not at the aggregate leadership score.

### 4.1. Perceptual Gaps in Competencies Across Roles

The largest manager–staff gaps appeared in Promoting Staff Retention, Recruiting Staff, Performing Supervisory Responsibilities, and Facilitating Staff Development—the very domains that structure day-to-day support, feedback, recognition, and growth. By contrast, the gap was comparatively smaller for Ensure Patient Safety and Quality, which tends to be standardized, audited, and jointly visible. International evidence mirrors this pattern: leaders frequently overestimate relational and developmental capabilities relative to staff appraisals, and precisely these behaviors are most consequential for intention to stay [1,7,15,16,17]. In this context, Wang et al. identified recruitment, supervision, and development as core competencies for nurse managers in Chinese tertiary hospitals, underscoring the salience of the same domains we observed as discrepant [2]. Jooste and Cairns reported misalignments between managers’ and nurses’ perceptions of self-leadership during capacity building, again pointing to role-based divergence in relational behaviors [17]. More recently, Long and Sochalski documented systematic discrepancies between supervisor self-evaluations and staff perceptions of leadership in healthcare teams, reinforcing the tendency toward managerial over-rating [18]; similar gaps emerged around performance appraisal processes in Moradi et al. [19]. Regionally, leadership practices in Saudi hospitals show variability and room for development in supervisory routines and staff engagement [20]; Al Mutair et al. found that quality of nursing work life—a determinant of retention—is tightly linked to day-to-day managerial behaviors [21]. These results are theoretically coherent: Katz’s human and conceptual skills, Transformational Leadership’s individualized consideration/coaching, and TPB’s attitudinal and control pathways all predict that when supervision and development feel thin or inconsistent, turnover cognitions rise.

### 4.2. Convergence on Turnover Intention Across Raters

Despite role differences in competency ratings, managers and staff did not differ in their perceptions of staff EMTIS (overall or subscales). This convergence is consistent with settings where shared unit realities—workload, acuity, staffing turbulence—anchor turnover cognitions for both leaders and staff [7,22,23]. Falatah’s integrative review of pandemic-era studies shows elevated turnover intention across roles when work conditions deteriorate [3], and Adhikari and Smith argue that global staffing pressures create system-level subjective norms that shape leaving/staying attitudes irrespective of position [22]. In Saudi Arabia, multicenter evidence links triggers such as workload and recognition to serious turnover intentions [24], and burnout dynamics described by Kelly et al. connect emotional strain to both organizational and positional turnover [16]. Convergence on the outcome lens, therefore, reduces concern about single-source bias (both raters assessed staff EMTIS) and shifts interpretive attention toward specific leadership enactments as plausible levers of change.

### 4.3. Why Domains Matter More than Composites

Small, inverse correlations clustered in Promoting Staff Retention, Perform Supervisory Responsibilities, Facilitate Staff Development, Ensure Patient Safety and Quality, and Lead Quality Improvement, whereas Managing Fiscal Planning and Facilitate Interpersonal, Group and Organizational Communication were weak or non-significant. This aligns with international syntheses showing that leadership–turnover links are modest overall but stronger when relational/developmental and safety-improvement behaviors are isolated [1,7,15,16,17]. In this context, Alsadaan et al. found that nurse leader behaviors tied to coaching, recognition, and performance support were associated with better staff outcomes [15]; and Goens & Giannotti concluded that transformational leadership is consistently linked to retention, primarily via staff development and engagement mechanisms [1]. Conversely, research indicates that fiscal and broad communication work influence intention indirectly—for example, through staffing fairness, workload, and perceived organizational support [13,16,25]. Pattali et al. demonstrated that the relationship between leadership style and turnover intention depends on perceived organizational support (moderation), illustrating why distal managerial work may need mediating pathways to affect EMTIS [25]. At the extreme, Ofei et al. showed that toxic leadership is strongly associated with turnover intention via job satisfaction (mediation) [23], highlighting the mechanism rather than the composite score per se. Methodologically, these patterns help explain our non-significant global NMCI–EMTIS result: aggregates dilute the contribution of the domains most proximal to EMTIS, and context-dependent pathways (rostering transparency, workload equity) attenuate the explanatory power of a single composite. Because EMTIS comprises Subjective Social Status, Organizational Culture, Personal Orientation, Expectations, and Career Development, the signal emerges most clearly through domain-to-facet pairing rather than composite-to-composite tests. Practically, this favors precision leadership: monitor and close domain-level gaps—especially in Promoting Staff Retention, Perform Supervisory Responsibilities, Facilitate Staff Development, Ensure Patient Safety and Quality, and Lead Quality Improvement—instead of relying on a global competency index [12,13,16,20,21,22,26].

### 4.4. Demographic Patterning and Organizational Context

Theory and prior evidence suggest that less-experienced and more highly educated nurses are more sensitive to shortfalls in Perform Supervisory Responsibilities and Facilitate Staff Development, and more likely to report higher turnover cognitions when those supports are weak [1,3]. Alilyyani et al. showed that leadership skill expectations increase with educational exposure, shaping perceptions of managerial adequacy in Saudi settings [27]. In the region, culturally adapted leadership programs (e.g., in Oman) improved competencies most relevant to engagement and retention, demonstrating how context-tuned curricula can close gaps where they matter operationally [26,27]. In practice, post-pandemic workload and multinational team structures necessitate explicit routines to: invite upward feedback safely; ensure equitable scheduling (a micro-behavior within Perform Supervisory Responsibilities); sustain recognition cadence (Promoting Staff Retention); and hold career-growth conversations (Facilitate Staff Development). Alharbi et al. documented variability in managers’ leadership practices in Saudi hospitals [20], while Jaber et al. identified burnout triggers and coping mechanisms that interact with leadership and workload, shaping turnover intention across facilities [28]. These contextual signals strengthen the case for unit-embedded, continuous leadership routines over one-off workshops.

### 4.5. Interpreting Magnitude and Setting Realistic Expectations

Effects were small but directionally consistent—typical of multi-determinant phenomena where staffing ratios, workload, and culture carry substantial weight [7,22,23,29]. Chang et al. synthesized turnover drivers and confirmed that leadership is a necessary but insufficient condition; durable impact emerges when leadership development is integrated with organizational levers such as staffing transparency and recognition systems [29]. Accordingly, organizations should expect incremental, cumulative gains from coaching rounds, recognition cadence, structured development plans, and transparent rostering, tracked with confidence intervals and retention KPIs to avoid over-promising on effect size.

### 4.6. Contribution and External Relevance

Beyond reaffirming that “leadership matters,” this study contributes two practice-ready elements. First, it offers a domain-level map of pressure points—Promoting Staff Retention, Perform Supervisory Responsibilities, Facilitate Staff Development, Ensure Patient Safety and Quality, Lead Quality Improvement—that aligns with international workforce directions emphasizing effective leadership, decent work, and retention [1,7,15,16,17,22,29]. Second, its dual-rater approach reduces single-source bias and renders manager–staff gaps auditable in routine dashboards—an approach compatible with the Saudi transformation agenda and adaptable to neighboring systems [5,6,20,21,22,23,25,28]. In this context, Sawafi et al. showed that region-specific leadership development can be culturally adapted without losing fidelity to core competencies, pointing to a feasible pathway for scaling precision-leadership programs in the Gulf [26].

### 4.7. Limitations

Despite the valuable insights gained from this study, several limitations must be acknowledged. First, the cross-sectional design constrains the ability to infer causality between nurse manager competencies, perceived competency gaps, and staff nurse turnover intentions. This design prevents the identification of changes over time, such as how perceptions of leadership or turnover intentions might evolve as nurses gain experience, encounter new organizational policies, or face prolonged crises. Future longitudinal studies could provide a more nuanced understanding of these dynamics, offering insight into how leadership perceptions and retention intentions fluctuate across different stages of professional development and organizational change. Additionally, quasi-experimental designs that intensively develop priority competency domains—**Promoting Staff Retention, Perform Supervisory Responsibilities, Facilitate Staff Development, and Ensure Patient Safety and Quality/Lead Quality Improvement—and then track both EMTIS and observed turnover would more directly test causal pathways (with a priori, preregistered hypotheses and fidelity checks) [20,21,23,26].

Second, the study relied on self-reported data, which may be subject to various biases, including social desirability or response bias. Staff nurses and nurse managers might have unintentionally over- or under-reported competencies or perceptions, potentially affecting the accuracy of the findings. In hierarchical, multinational teams, cultural response styles (e.g., acquiescence, deference) may also shape ratings. Future research could address this limitation by incorporating mixed methods approaches, such as structured interviews, focus groups, or observational assessments, to validate and enrich self-reported data. Our dual-rater design (manager self-ratings and staff ratings of the same managers) partially mitigates single-source bias for competencies and, importantly, both groups rated the same outcome lens (staff EMTIS); nonetheless, role-lens variance remains possible and should be examined with measurement-invariance testing across raters and triangulation with administrative data (staffing rosters, vacancy/turnover logs, and exit interviews) [12,15].

Third, the study was conducted in two tertiary governmental hospitals located in Saudi Arabia, which may limit the generalizability of the findings. While these hospitals represent significant clinical settings and provide insight applicable to similar healthcare environments in the region, the results may not fully reflect dynamics in smaller facilities, private hospitals, or healthcare systems in other countries. Moreover, the use of a quota/convenience approach and the potential for nonresponse bias may constrain the representativeness of the accessible population. Expanding the geographic and institutional scope in future research would enhance generalizability and provide broader insights into how organizational structure, resource availability, and regional healthcare policies influence competency gaps and turnover intentions. Multi-site, multi-sector, and multi-region sampling with multilevel models (e.g., unit- and hospital-level effects) can clarify contextual influences on leadership-turnover links [16,20,21,25,26,29,30].

Finally, while the study focused on nurse manager competencies and selected demographic characteristics, other potentially influential factors were not examined in depth. Variables such as organizational culture, staffing ratios, workload, support systems, perceived organizational support, and external stressors may interact with perceived competency gaps, shaping turnover intentions and staff satisfaction. Future research should explore these additional factors to provide a more holistic understanding of the determinants of nurse retention and workforce stability, particularly in high-pressure or crisis-prone healthcare settings. Analytically, mediation pathways should be probed to connect distal managerial work (e.g., Managing Fiscal Planning) with proximal staff experiences (e.g., staffing/scheduling fairness) and EMTIS; moderation by experience and education should be tested explicitly [13,16]. Given that effect sizes were small but consistently patterned across EMTIS facets, adequately powered samples, confidence-interval-based monitoring, and repeated measures will be essential to detect meaningful, domain-specific changes over time [20,21,26].

### 4.8. Implications and Recommendations

The findings of this study have important implications for nursing practice, healthcare management, education, and future research. They highlight the critical role of nurse leadership competencies in shaping workforce stability and the necessity of addressing gaps to enhance staff retention and patient care quality. Because managers and staff converged on perceived staff turnover intention while diverging on specific competency domains, organizations should prioritize domain-specific leadership improvement (Promoting Staff Retention, Perform Supervisory Responsibilities, Facilitate Staff Development, Ensure Patient Safety and Quality, Lead Quality Improvement) and de-emphasize reliance on global composite scores that may mask actionable signals [13,16,20,21,25,26]. Consistent with the theory-driven hypotheses, improvement efforts should explicitly use a dual-rater (“dual-lens”) approach to monitor manager–staff gaps by NMCI domain and link “gap-closure” to EMTIS facets and retention KPIs over time.

#### 4.8.1. Implications for Nursing Practice

Healthcare organizations should implement structured, competency-based leadership development programs that target the domains identified as gaps in this study, particularly Recruit Staff, Promoting Staff Retention, and Perform Supervisory Responsibilities. These programs should be tailored to different staff groups, recognizing that novice and highly educated nurses may be more sensitive to leadership deficiencies. Operationalize this targeting with dual-lens diagnostics (manager self- vs. staff ratings) and track domain-level gap-closure with confidence intervals and run-charts, linking progress to EMTIS facets and retention KPIs on service dashboards [20,21,26].Feedback-informed leadership evaluation systems can be instituted, whereby staff nurses’ perceptions are systematically collected and used to guide managerial development. This approach ensures that leadership improvements are aligned with the actual needs and expectations of the nursing workforce. Given the comparatively smaller manager–staff gap on Ensure Patient Safety and Quality, organizations can leverage existing policy/audit routines in this domain as a platform to spread alignment practices to perform Supervisory Responsibilities and Facilitate Staff Development [12,13]. Embed a quarterly cadence (audit → feedback → micro-skill practice → re-audit) to sustain behavior change.Nurse managers should engage in continuous professional development focused on both technical skills (operational and clinical competencies) and relational skills (communication, mentoring, and motivational leadership), which are essential for fostering a supportive work environment and enhancing nurse satisfaction. Micro-skills should include equitable scheduling checks (Perform Supervisory Responsibilities), structured one-to-one coaching, recognition cadence, and career-growth conversations (Facilitate Staff Development)—behaviors most closely associated with lower turnover cognitions in this study [16,20,21,25,26]. Where feasible, align rostering practices and recognition with transparent criteria to strengthen perceived fairness.

#### 4.8.2. Implications for Healthcare Policy

Policymakers can leverage these findings to establish national competency frameworks for nurse managers, integrating standardized assessment, evaluation, and certification processes. Such frameworks will help ensure leadership quality across healthcare institutions and support workforce sustainability in the long term. Frameworks should weight domains with demonstrated relevance to turnover cognitions (Promoting Staff Retention, Perform Supervisory Responsibilities, Facilitate Staff Development, Ensure Patient Safety and Quality, Lead Quality Improvement) and require dual-lens measurement as part of accreditation and performance review [20,21,26,30]. Publish unit-level gap metrics with longitudinal CIs to encourage transparent improvement.Retention strategies should be context-specific, recognizing differences in workforce demographics. For instance, targeted mentorship and recognition programs for less experienced nurses or newly qualified staff may mitigate turnover risks in critical care and high-acuity units. At the policy level, mandate reporting of manager-staff gap metrics (by NMCI domain) and tie improvement to incentives (e.g., recognition funds, development time), thereby aligning governance with workforce sustainability goals [20,21,26]. Complement leadership policy with organizational levers (staffing transparency, workload management, career pathways) so distal work (e.g., Managing Fiscal Planning) influences proximal experiences (fair rosters, support).

#### 4.8.3. Implications for Nursing Education

Nursing curricula should incorporate leadership development modules that prepare future nurse managers for the challenges of contemporary healthcare environments, emphasizing practical skills in supervision, retention strategies, and staff recruitment. Integrate 360-style feedback and dual-lens audits early so leaders learn to calibrate self-perceptions with staff perceptions and close gaps proactively. [12,13,20]Simulation-based and scenario-driven training can provide prospective nurse managers with opportunities to practice decision-making, conflict resolution, and team leadership in safe and controlled environments, thereby bridging the gap between theory and real-world practice. Scenario design should mirror the micro-behaviors linked to EMTIS facets (e.g., coaching a struggling staff nurse, redesigning schedules for fairness, leading a safety huddle) and include structured debriefs tied to specific NMCI domains [16,20,21,25,26]. Assess performance with standardized rubrics mapped to NMCI domains and provide targeted feedback on gap areas.

#### 4.8.4. Implications for Future Research

Future studies should explore the moderating effects of demographic characteristics (age, experience, and education) through formal moderated regression analyses to understand how these factors influence the relationship between perceived leadership competency gaps and turnover intention. Cross-level moderation (unit, service line) should be examined using multilevel models to capture contextual influences [16,20,21,25,26,29,30].Longitudinal studies are recommended to examine changes in competency perceptions and turnover intention over time, particularly in the context of healthcare transformations or crises such as pandemics. Quasi-experimental trials that intensively develop specific competencies (e.g., supervision or staff development) can test whether improvements produce downstream reductions in EMTIS and observed turnover [20,21,23,26]. Pre-registration of domain-specific hypotheses, fidelity monitoring, and minimum detectable-effect power calculations are recommended.Comparative studies across multiple countries or regions could investigate cultural and organizational influences on leadership perception, competency gaps, and retention outcomes, providing insights for global nursing workforce planning. Future work should also test mediating pathways (e.g., fiscal planning → staffing/scheduling fairness → EMTIS) and assess measurement invariance across rater groups to rule out artifactual differences [13,16]. Triangulate survey data with administrative sources (rosters, vacancies, exits) to strengthen inference.

#### 4.8.5. Actionable Recommendations

Prioritize domain-specific development in Promoting Staff Retention, Perform Supervisory Responsibilities, Facilitate Staff Development, Ensure Patient Safety and Quality, Lead Quality Improvement; avoid over-reliance on global composites. [16,20,21,25,26]Implement quarterly dual-lens audits (manager self vs. staff ratings) with explicit gap-closure thresholds; display CIs and run-charts; review alongside EMTIS facets and retention KPIs [20,21,26].Tailor retention strategies, workforce mix (novice vs. experienced; advanced-degree cohorts): differentiate coaching content and intensity; pair novices with trained preceptors; align recognition to transparent criteria [1,3,21].Couple leadership routines with organizational levers (staffing transparency, equitable rostering, recognition/progression policies) so leadership effects translate into proximal experiences that matter for EMTIS [13,16,25,29]Institutionalize a learn-and-improve loop (audit → feedback → micro-skill practice → follow-up audit), embedded in unit quality improvement initiatives, and report progress at governance forums to sustain accountability and focus [20,21,26].

## 5. Conclusions

This study identifies modest but consistent, domain-specific associations between nurse-manager competencies and staff nurses’ turnover intention. The largest perceptual gaps clustered in Promoting Staff Retention, Recruit Staff, Perform Supervisory Responsibilities, and Facilitate Staff Development, and small inverse associations with EMTIS were most evident for Promoting Staff Retention, Perform Supervisory Responsibilities, Facilitate Staff Development, Ensure Patient Safety and Quality, and Lead Quality Improvement.

Furthermore, demographic characteristics such as age, experience, and education may shape staff sensitivity to these gaps, suggesting the need for tailored leadership interventions. Notably, manager–staff divergence was smaller in ensuring patient safety and quality, whereas the most consequential signals for turnover cognitions were domain-specific (Promoting Staff Retention; Perform Supervisory Responsibilities; Facilitate Staff Development; Ensure Patient Safety and Quality; Lead Quality Improvement). The global competency–turnover association was non-significant, underscoring the value of granular, domain-level measurement and improvement.

Addressing these competency gaps through targeted, evidence-based leadership development programs can enhance staff retention, improve team cohesion, and strengthen workforce sustainability. These interventions are especially important in high-pressure healthcare settings and during periods of systemic transformation, such as those observed in the post-pandemic era. Given the small but patterned effects observed, organizations should favor continuous, unit-embedded routines—such as dual-lens audits (manager self-ratings vs. staff ratings) by NMCI domain, explicit gap-closure targets with confidence-interval monitoring, and linkage to EMTIS facets and retention KPIs—over one-off trainings. Where appropriate, couple leadership routines with organizational levers (staffing transparency, equitable rostering, recognition, and career pathways) so improvements translate into proximal experiences that influence EMTIS.

The findings have both regional and global relevance, informing strategies to optimize nurse leadership effectiveness, reduce turnover, and maintain high-quality patient care. Future research should pre-register domain-specific hypotheses and explore longitudinal changes in perceptions of managerial competencies, examine additional organizational and environmental factors, and evaluate the effectiveness of tailored interventions to ensure the long-term stability and resilience of the nursing workforce. Priority next steps include quasi-experimental evaluations of targeted competency development, multilevel analyses across diverse settings, measurement-invariance tests across raters, and mediation models that connect distal managerial work (e.g., Managing Fiscal Planning) to proximal staff experiences (e.g., staffing/scheduling fairness) and retention outcomes.

### What This Paper Added to the Literature/Field

Introduces a dual-rater, single-outcome (EMTIS) design that reduces single-source bias while revealing auditable manager–staff perceptual gaps across NMCI domains.Demonstrates that domain-specific (micro-competency) signals—not global composites—are the actionable correlates of turnover intentions, consistent with small but patterned effects.Identifies priority NMCI domains for precision leadership: Promoting Staff Retention; Perform Supervisory Responsibilities; Facilitate Staff Development; Ensure Patient Safety and Quality; Lead Quality Improvement.Shows convergence of manager and staff perceptions on EMTIS, focusing attention on leadership enactment rather than rater artifact.Provides a practice-ready framework- dual-lens diagnostics by domain, gap-closure targets with CI monitoring, and retention-linked dashboards compatible with regional transformation agendas and international workforce directions.

## Figures and Tables

**Figure 1 healthcare-13-02506-f001:**
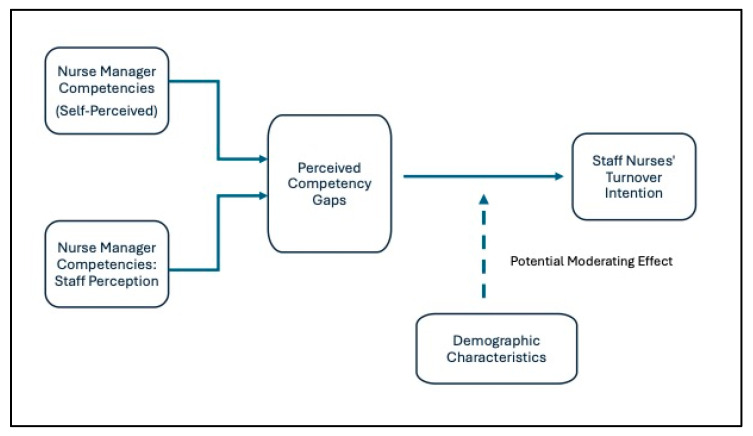
Conceptual framework illustrating competencies, perceptual gaps, turnover intention, with demographics as potential moderators.

**Table 1 healthcare-13-02506-t001:** Selected Participant Characteristics (n = 396).

Characteristic	Category	(Staff Nurse) n (%)	(Nurse Manager) n (%)
Age	<30	99 (44.0)	10 (5.8)
	30 > 35	56 (24.9)	28 (16.4)
	35 > 40	24 (10.7)	39 (22.8)
	40 > 45	22 (9.8)	49 (28.7)
	≥45	24 (10.7)	45 (26.3)
X (SD)		35.6 (6.8)	42.3 (7.5)
Education	Bachelor	124 (55.1)	56 (32.7)
	Diploma	87 (38.7)	1 (0.6)
	Master+ *	14 (6.2)	114 (66.7)
Exp	1 > 5	81 (36.0)	11 (6.4)
	5 > 10	75 (33.3)	42 (24.6)
	10 > 15	69 (30.7)	118 (69.0)
	≥15	0 (0.0)	0 (0.0)
Gender	Female	123 (54.7)	84 (49.1)
	Male	102 (45.3)	87 (50.9)
Marital Status	Single	112 (49.8)	38 (22.2)
	Married	84 (37.3)	84 (49.1)
	Divorced/Widowed	29 (12.9)	49 (28.7)

Notes: * Master+ (Master’s/Doctorate).

**Table 2 healthcare-13-02506-t002:** Comparison of Nurse Managers’ Self-Perceptions and Staff Nurses’ Perceptions.

Nurse Manager’s Competencies Subscales	Mean (SD)				F	*p* Value *
	N. Manager		S. Nurse			
	M (SD)	95% CI	M (SD)	95% CI		
Promoting Staff Retention	4.00 (0.75)	[3.89, 4.11]	2.98 (0.85)	[2.87, 3.09]	0.36	0.84
Recruit Staff	3.98 (0.73)	[3.87, 4.09]	3.00 (0.86)	[2.89, 3.11]	0.05	0.94
Facilitate Staff Development	4.07 (0.72)	[3.96, 4.18]	3.43 (0.87)	[3.32, 3.54]	0.25	0.87
Perform Supervisory Responsibilities	4.08 (0.70)	[3.98, 4.18]	3.23 (0.80)	[3.13, 3.33]	0.46	0.51
Ensure Patient’s Safety and Quality Care	3.90 (0.76)	[3.79, 4.01]	3.75(0.85)	[3.64, 3.86]	0.32	0.85
Conduct Daily Unit Operations	4.00 (0.71)	[3.89, 4.11]	3.32 (0.75)	[3.22, 3.42]	0.16	0.90
Manage Fiscal Planning	4.10 (0.69)	[4.00, 4.20]	3.45 (0.80)	[3.35, 3.55]	0.15	0.69
Facilitate Interpersonal, Group, and Organizational Communication	4.00 (0.71)	[3.89, 4.11]	3.34 (0.81)	[3.23, 3.45]	0.49	0.48
Lead Quality Improvement	3.90 (0.81)	[3.78, 4.02]	3.28 (0.80)	[3.18, 3.38]	0.33	0.65
Overall NMCI	3.67 (0.61)	[3.58, 3.76]	3.04 (0.74)	[2.94, 3.14]	0.114	0.73

* *p* value ≤ 0.05.

**Table 3 healthcare-13-02506-t003:** Comparison Between Staff Nurses and Nurse Managers in Relation to EMTIS Turnover Intention.

Turnover Intention Subscales	Mean (SD)				F	*p*-Value *
	Nurse Manager		Staff Nurses			
	M (SD)	95% CI	M (SD)	95% CI		
Subjective Social Status	2.34 (1.20)	[2.16, 2.52]	2.32 (1.05)	[2.18, 2.46]	15.17	0.334
Organizational Culture	2.87 (1.90)	[2.59, 3.15]	2.72 (1.25)	[2.56, 2.88]	13.43	0.293
Personal Orientation	3.00 (1.40)	[2.79, 3.21]	2.88 (1.28)	[2.71, 3.05]	14.76	0.362
Expectations	2.98 (1.85)	[2.70, 3.26]	2.89 (1.48)	[2.70, 3.08]	15.11	0.281
Career Development	3.00 (1.44)	[2.78, 3.22]	2.83 (1.69)	[2.61, 3.05]	23.21	0.621
Overall	3.16 (1.28)	[2.97, 3.35]	3.00 (1.15)	[2.85, 3.15]	21.32	0.173

* *p* < 0.05.

**Table 4 healthcare-13-02506-t004:** Correlation Between Nurse Manager Competencies and Staff Nurse Turnover Intention.

Turnover Subscales	Managers’ Competencies Subscales								
	Promoting Staff Retention	Recruited Staff	Facilitate Staff Development	Perform Supervisory Responsibilities	Ensure Patient Safety and Quality	Conduct Daily Unit Operations	Managing Fiscal Planning	Facilitate Interpersonal, Group and Organizational Communication	Lead Quality Improvement
Subjective Social Status	r = −0.15 * *p* = 0.002	r = −0.14 * *p* = 0.007	r = −0.14 * *p* = 0.005	r = −0.15 * *p* = 0.005	r = −0.12 * *p* = 0.01	r = −0.09 *p*= 0.06	r= −0.05 *p* = 0.24	r = −0.06 *p* = 0.23	r = −0.15 * *p* = 0.003
Organizational Culture	r = −0.12 * *p* = 0.01	r = −0.13 * *p*= 0.01	r = −0.14 * *p* = 0.02	r = −0.13 * *p* = 0.01	r = −0.11 * *p* = 0.03	r = −0.10 *p* = 0.05	r = −0.03 *p* = 0.58	r = −0.05 *p* = 0.42	r = −0.13 * *p* = 0.01
Personal Orientation	r = −0.13 * *p* = 0.01	r = −0.13 * *p* = 0.008	r = −0.12 * *p* = 0.01	r = −0.14 * *p* = 0.007	r = −0.14 * *p* = 0.007	r = −0.11 * *p* = 0.02	r = −0.05 *p* = 0.30	r = −0.04 *p* = 0.33	r = − 0.12 * *p* = 0.02
Expectations	r = −0.13 * *p* = 0.01	r = −0.14 * *p* = 0.005	r = −0.18 * *p* = 0.01	r = −0.14 * *p* = 0.006	r = −0.12 * *p* = 0.01	r = −0.10 * *p* = 0.03	r = −0.04 *p* = 0.40	r = −0.07 *p*= 0.16	r= −0.13 * *p* = 0.008
Career Growth	r = −0.19 * *p* = 0.01	r = −0.13 * *p* = 0.01	r = −0.12 * *p* = 0.02	r = −0.13 * *p* = 0.01	r = −0.13 * *p* = 0.008	r = −0.08 *p* = 0.09	r = −0.05 *p*= 0.31	r = −0.08 *p* = 0.14	r = −0.13 * *p* = 0.01
	Total Nurse Manager Competencies								
Overall Turnover Intention	r = −0.077 *p* = 0.124								

* *p* < 0.05.

## Data Availability

The data that support the findings of this study are available from the corresponding author upon reasonable request. The data are not publicly available due to privacy restrictions.

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
