# Peer review of "Bridging Leadership Competency Gaps and Staff Nurses’ Turnover Intention: Dual-Rater Study in Saudi Tertiary Hospitals"

_healthcare, 2025, doi:10.3390/healthcare13192506_

Round 1
Reviewer 1 Report
Comments and Suggestions for Authors
The article addresses a highly relevant topic. The text is well structured, but it requires adjustments to achieve good scientific quality, especially strengthening the justification, detailing methods, expanding results, and deepening the discussion in terms of originality and articulation with global policies.
Title
The term “Insights from a Cross-Sectional Study” is generic. I suggest specifying the context more precisely, for example: “… A Cross-Sectional Study in Tertiary Hospitals in Saudi Arabia”.
Abstract
-
Present the objective according to the wording in item 1.1.
-
Insert the data collection period.
-
In the results, include statistical data.
Keywords
They are coherent with the title and objective but need adjustments to ensure fidelity to the MeSH thesaurus.
Introduction
-
Insert references for lines 44–46.
-
Which references support the statement in lines 53–55?
-
The text does not contextualize what is already known about nurse and nurse manager competency gaps.
-
The knowledge gap is implicit but should be made more explicit.
-
Insert references for the statements in lines 80–82 and 87–89.
Methods
-
Insert the study period.
-
In Table 1, include an explanatory note about the meaning of the education levels: Bachelor/Diploma.
-
For the variable Years of Experience, please present the previous categories. It seems they were categorized in 5-year intervals, but note: where does the number 15 fit? Suggested categorization: 1–5; 6–10; 11–15; 16 or more.
-
The sample selection is not sufficiently detailed (convenience sampling? random?). There is no mention of whether a prior sample size calculation was performed.
-
How many times were eligible participants contacted to join the study?
-
Insert information about the response rate.
-
Were the instruments described in items 2.3.1 and 2.3.2 culturally validated for the Saudi Arabian population?
-
Line 161: which was the secure online platform?
Results
-
The Nurse Manager Competency Inventory has 11 domains, but Table 2 presents only 3. Even if the remaining domains were not statistically significant, they are still important results and contribute to strengthening the evidence on this topic.
-
The text in lines 208–211 and 217–219 represents an interpretation of the results and should therefore be moved to the discussion section.
-
The Expanded Multidimensional Turnover Intention Scale (EMTIS) has 5 domains; thus, the authors should present results for all of them.
Discussion
The discussion is superficial, mainly reinforcing findings already known; it lacks emphasis on the study’s original contribution and dialogue with global policies such as the WHO Global Strategic Directions for Nursing and Midwifery and the ICN Renewing the Definitions of ‘Nursing’ and ‘a Nurse’. The international implications should be expanded beyond the Saudi context.
Author Response
Thank you very much for taking the time to review this manuscript. Please find the detailed responses and the corresponding revisions/corrections in the attached file, and the tracked changes in the resubmitted files.

Reviewer 2 Report
Comments and Suggestions for Authors
Dear Authors,
Thank you very much for allowing me to review your article. The manuscript addresses a highly relevant topic in the current context of nursing human resource management: the relationship between nurse managers’ competencies, staff perceptions, and turnover intention. The focus on Saudi Arabia, in the framework of healthcare system transformation, provides added value, while the use of a dual-rater design constitutes a novel contribution. Below, I detail strengths, weaknesses, and recommendations by section.
Title and Abstract
The title is clear, specific, and accurately reflects the purpose of the research. The abstract summarizes objectives, methodology, main findings, and practical contributions effectively, facilitating the reader’s understanding. However, the abstract provides minimal statistical detail, which makes it difficult to assess the true magnitude of the effects. Furthermore, the emphasis on small associations should be tempered to avoid generating excessive expectations about the impact of the findings.
Suggestions for improvement: I recommend including the key limitations of the study in the abstract, as they are critical for interpreting the validity of the results. It would also be helpful to highlight more clearly the international relevance of the study to increase its appeal to audiences not directly linked to the Saudi context.
Introduction
The introduction adequately presents the global problem of nursing shortages and situates it within the transformation of the Saudi healthcare system. It integrates a solid theoretical framework (three complementary theories: Katz, Transformational Leadership, and the Theory of Planned Behavior). It also identifies a clear knowledge gap: differences in perceptions between managers and nurses in the Middle Eastern context. However, the introduction is somewhat repetitive in places. While the importance of demographic factors is mentioned, their relevance could be developed further.
Suggestions for improvement: Streamline the narrative to avoid redundancies and make the text more concise. Ideally, explicit hypotheses derived from the theoretical framework should be formulated, as this would reinforce the coherence between theory, research questions, and results.
Objective and Research Questions
The overall objective and the research questions are well structured and directly linked to the identified gap. However, the objective is broad; it would have been preferable to prioritize the most relevant questions (e.g., the relationship between competencies and turnover) and treat the others as secondary analyses.
Suggestions for improvement: Reformulate the objective in more analytical terms, specifying clearly which associations or effects were intended to be tested.
Methodology
The methodology adopted is consistent with the exploratory nature of the study and rests on a reliable foundation. The cross-sectional, descriptive-correlational design is appropriate for detecting basic differences and associations, and the inclusion of two hospitals with a large sample (n = 396) adds robustness. The dual-rater strategy is one of the most original and valuable features, as it reduces single-source bias and allows for perception comparisons. The instruments used (NMCI and EMTIS) have been widely validated and show high internal consistency (α = 0.93), which ensures reliability. In addition, compliance with ethical standards is thoroughly detailed and well documented.
The main methodological limitations lie in the scope of the analyses. The exclusive use of ANOVA and bivariate correlations limits explanatory depth and leaves unexamined the role of demographic variables, which do appear in the conceptual framework. This creates a slight inconsistency between the theoretical proposition and the empirical execution. While the authors chose not to apply multivariate models, there are strategies for improvement within the same approach: report effect sizes (Cohen’s d, partial η²) and confidence intervals to better capture the magnitude of the differences; explore the distribution of gaps (e.g., percentage of discrepancies ≥ 1 point between self-assessment and staff evaluation) to add descriptive evidence; conduct stratified analyses by age, education, or experience, which would enrich the moderation analysis without requiring advanced models; and report internal consistency of each subscale in the current sample to confirm reliability in this context. With these improvements, the chosen cross-sectional design would gain strength and provide greater interpretive clarity without departing from the declared methodology.
Results
The results are presented in an orderly manner, with clear tables and appropriate comparisons. It is positive that the authors transparently indicate the lack of statistical significance in some analyses. However, too much emphasis is placed on descriptive differences without statistical support, and effect sizes, which are modest, are interpreted as having greater practical significance than they actually do. To strengthen this section, it would be important to report confidence intervals alongside means and to temper the interpretation of non-significant results.
Discussion
The discussion integrates the findings with international and regional literature and correctly highlights the importance of specific competencies in retention, development, and supervision. However, the practical relevance of the findings is somewhat overstated, despite the small effect sizes. Moreover, the absence of significant associations at the global level is mentioned but not explored in sufficient depth; reflecting on this would enrich the theoretical contribution. Additionally, while contextual factors such as the pandemic or healthcare transformations are mentioned, their interaction with the perceptions observed is not sufficiently analyzed. A deeper reflection on the role of Saudi organizational culture and its potential influence on leadership perceptions would also strengthen the discussion.
Limitations, Implications, and Conclusions
The manuscript acknowledges the main methodological limitations: cross-sectional design, self-report, and geographic restriction. This recognition is commendable, though consideration of social desirability bias and cultural influences on leadership perceptions is lacking. Practical and policy implications are well articulated, but in some cases they appear aspirational and not directly derived from the data. It would be advisable to differentiate more clearly between what the results support and what constitutes a normative projection—i.e., ambitious recommendations that go beyond what was empirically tested. The conclusions are coherent, though they convey an overly optimistic tone in relation to the modesty of the observed effects. A more nuanced writing would reinforce credibility.
Overall Assessment
Overall, this study represents a valuable contribution by addressing a genuine knowledge gap in the Middle Eastern context through an innovative dual-rater design and the use of reliable instruments. The methodology adopted has sufficient reliability for an exploratory cross-sectional study, and the results could be significantly improved with more comprehensive descriptive analysis, the presentation of effect sizes and confidence intervals, and subgroup analyses that do not require changing the methodological approach. The manuscript has potential but requires substantial revision to better align the conceptual framework with the analysis, temper the conclusions according to the actual strength of the findings, and reinforce the statistical presentation.
Author Response

(The authors gave the same response as above.)

Round 2
Reviewer 1 Report
Comments and Suggestions for Authors The modifications made improved the quality of the manuscriptReviewer 2 Report
Comments and Suggestions for Authors
Dear authors,
Thank you very much for this second version of the article. I would like to sincerely thank you for the thoughtful and thorough way in which you have addressed all of the comments. The revised manuscript reflects significant improvements in several key areas.
First, the title and abstract are now more informative and balanced. By adding effect sizes, confidence intervals, and key limitations, you have greatly improved transparency and ensured that the modest nature of the associations is clearly conveyed. The inclusion of a note on the study’s international relevance also broadens its appeal beyond the Saudi context.
Second, the introduction and theoretical framework have been strengthened. The narrative is more concise and avoids redundancies, while the addition of explicit hypotheses (H1–H4) provides a clearer bridge between the theoretical foundation, the research aims, and the empirical analyses. Expanding the rationale for demographic factors such as education, experience, and age adds further depth and aligns well with the exploratory analyses reported later.
Third, the objectives and research questions are now formulated in more analytical terms. By prioritizing domain-specific NMCI–EMTIS associations as the primary aim and treating cross-role comparisons and demographic influences as secondary, you have enhanced the focus and coherence of the study.
Fourth, the methodology and analysis have been meaningfully enhanced while remaining within the original cross-sectional design. Reporting effect sizes (Cohen’s d, partial η²), 95% confidence intervals, and the percentage of ≥1-point discrepancies adds clarity and robustness. Stratified descriptive analyses by age, education, and experience, as well as reporting Cronbach’s α for each subscale in the current sample, strengthen the reliability and interpretive depth of the findings. The decision to leave multivariate modeling for future research is clearly justified, and the adjustments made already enrich the contribution of the current design.
Fifth, the results section is now presented with greater methodological rigor and neutrality. The inclusion of confidence intervals alongside group means, and the explicit reporting of non-significant findings, prevents overinterpretation. I appreciate that interpretive claims have been reserved for the discussion, ensuring that results are communicated in a transparent and non-speculative manner.
Sixth, the discussion has been recalibrated effectively. The interpretation of findings now reflects the modest effect sizes more faithfully, avoiding overstatement. The additional explanation for the lack of global associations—such as heterogeneity across domains and the multifaceted nature of EMTIS—adds theoretical depth. Expanding the analysis of contextual factors, including post-pandemic pressures and the Vision 2030 transformation, as well as the reflection on Saudi organizational culture and hierarchical dynamics, provides important cultural grounding and enhances the international relevance of the study.
Finally, the sections on limitations, implications, and conclusions are much improved. The manuscript now acknowledges social desirability bias and cultural response styles, along with role-based variance, which increases transparency. The distinction between recommendations directly supported by the data (e.g., domain-level signals, dual-lens audits) and broader policy options is very welcome, as it clarifies the scope of the evidence and reduces the risk of overstating practical implications. The conclusions are now written in a more measured and credible manner, reflecting the modest yet consistent domain-level associations observed.
In sum, these revisions substantially strengthen the manuscript. They improve its theoretical–empirical alignment, increase methodological transparency, and situate the findings more convincingly in both local and international contexts. The paper now represents a credible, balanced, and valuable contribution to the literature on nursing leadership and workforce management.